

# Insights from single-strain and mixed culture experiments on the effects of heatwaves on freshwater flagellates

Lisa Boden[1], Chantal Klagus[1] and Jens Boenigk[1,2]

[1] Department Biodiversity, University of Duisburg–Essen, Essen, North Rhine Westphalia, Germany
[2] Center for Water and Environmental Research, University of Duisburg–Essen, Essen, North Rhine Westphalia, Germany

## ABSTRACT

The increasing frequency and intensity of heatwaves driven by climate change significantly impact microbial communities in freshwater habitats, particularly eukaryotic microorganisms. Heterotrophic nanoflagellates are important bacterivorous grazers and play a crucial role in aquatic food webs, influencing the morphological and taxonomic structure of bacterial communities. This study investigates the responses of three flagellate taxa to heatwave conditions through single-strain and mixed culture experiments, highlighting the impact of both biotic and abiotic factors on functional redundancy between morphologically similar protist species under thermal stress. Our results indicate that temperature can significantly impact growth and community composition. However, density-dependent factors also had a significant impact. In sum, stabilizing effects due to functional redundancy may be pronounced as long as density-dependent factors play a minor role and can be overshadowed when flagellate abundances increase.

## INTRODUCTION

Microbial communities frequently experience and cope with natural fluctuations in abiotic environmental conditions (*Ruokolainen et al., 2009*). However, strong changes caused by climate change and anthropogenic activities put ecosystems under severe pressure (*IPBES, 2019*). Microorganisms have evolved mechanisms to withstand changing environmental conditions (*Bernhardt et al., 2020*), but tolerances to abiotic stressor vary strongly (*Díaz-Almeyda et al., 2017*; *Mascarin et al., 2018*; *Jain & Saraf, 2021*). Consequently, abiotic disturbances can result in lower abundance of individual species in the microbial community or total loss of species (*Schulhof et al., 2020*; *Proesmans et al., 2022*). Such changes in biodiversity can significantly impact ecosystem functions and food webs (*Mooney et al., 2009*; *IPBES, 2019*). However, the impact of losing individual species can be lowered by the presence of other species in the community that perform the same function, an attribute known as functional redundancy (*Walker, 1992*; *Naeem, 1998*). Functional redundancy is hypothesized to enhance the resilience and stability of ecosystems during

Corresponding author
Lisa Boden, lisa.boden@uni-due.de

the occurrence of stressors by maintaining ecological functions despite changes in taxon composition (*Biggs et al., 2020*; *Chen et al., 2022*). Here we address functional redundancy between cryptic species and its changing significance in the light of density-dependent and density-independent factors—an aspect about which, to our knowledge, little is known, particularly in freshwater habitats.

As a major abiotic stressor, increased temperatures threaten biodiversity across different habitats (*Urrutia-Cordero et al., 2017*; *Maire et al., 2022*; *Lau et al., 2024*). In addition to the long-term increase of the average global surface temperature, more frequent and intense heatwaves have been predicted as a consequence of climate change (*Woolway et al., 2021*; *IPCC, 2021*). Temperature thresholds and the duration of heatwaves are region-specific and cannot be universally defined (*Xu et al., 2016*). However, heatwaves are generally characterized by relatively short but intense increases in temperature, representing one of the most significant challenges in aquatic habitats (*Sun & Arnott, 2022*). Climate change scenarios predict that temperatures during heatwaves in freshwater bodies can increase by up to 4 °C for several days (*Woolway et al., 2021*). During the occurrence of a heatwave, organisms are rapidly pushed beyond the limit of their temperature tolerances, which may affect community composition and ecosystem functions more profoundly than a gradual long-term temperature increase (*Vasseur et al., 2014*; *Stillman, 2019*). Heatwaves have been shown to cause biodiversity loss in both marine and freshwater habitats (*Brauko et al., 2020*; *Sabater et al., 2022*). Particularly for eukaryotic microorganisms, heat stress has resulted in significant shifts in community composition (*Hao et al., 2018*; *Thomson & Manoylov, 2019*). Interestingly, it is assumed that short-term disturbances like heatwaves are succeeded by complete or partial recovery (*Bender, Case & Gilpin, 1984*; *Harris et al., 2018*).

Colorless heterotrophic nanoflagellates play an important role in regulating the quantity and biomass of bacterial populations and profoundly influence the morphological and taxonomic structure of bacterial communities (*Sherr & Sherr, 2002*). Consequently, alterations in both the quantity and quality of bacterivory, resulting from shifts in the flagellate community, can influence the overall structure of aquatic food webs. The Chyrosphytes are among the most important bacterivorous grazers in freshwater habitats (*Finlay & Esteban, 1998*). They are widely distributed across various habitats, primarily inhabiting freshwaters but also extending into terrestrial and marine ecosystems (*Kristiansen & Škaloud, 2017*). They further encompass a broad range of different morphologies (*Škaloud, Kristiansen & Škaloudová, 2013*; *Skaloudova & Skaloud, 2013*). The morphology of flagellates has been shown to correlate with various aspects of predator–prey interactions, including preferences for food size and feeding mechanisms (*Boenigk & Arndt, 2000a*; *Boenigk & Arndt, 2000b*). Consequently, flagellates exhibiting similar morphologies can feed on the same prey, thereby serving equivalent roles in the food web. Interestingly, molecular analysis demonstrated high molecular diversity behind distinct morphological forms (*Grossmann et al., 2016*). This cryptic diversity likely resulted from parallel evolution, wherein similar morphological forms evolved independently in different lineages (*Graupner et al., 2018*). The ecological significance of this phenomenon, however, remains unclear. In the order Ochromonadales, significant differences in temperature tolerance were observed

among clades with similar morphologies (*Boenig et al., 2007*; *Nolte et al., 2010*). Such clade-specific responses to changing environmental conditions can result in a shift of the community composition (*Boden, Sieber & Boenigk, 2023*). This suggest that cryptic diversity might disguise a turn-over of cryptic taxa that are differently adapted to abiotic factors, thereby stabilizing the structure of food webs and ecosystem functions during the occurrence of stressors despite significant changes in the taxon composition (*Boden, Sieber & Boenigk, 2023*).

Here, we investigate the effects of heatwaves on the growth of three cryptic taxa associated with *Pedospumella*, *Spumella*, and *Poteriospumella* to gain a better understanding of the ecological importance of cryptic diversity and its potential impact on the stability of ecosystems when facing abiotic disturbances. Given the variation in temperature tolerance, we anticipate that the three strains will respond differently to the temperature increase. Additionally, we expect the strains to grow differently in mixed cultures compared to the single-strain cultures because biotic interactions between the strains will influence the growth of each taxon. We predict more pronounced fluctuations in the total abundance of flagellates in single-strain cultures compared to mixed cultures because the presence of multiple taxa with varying temperature tolerances in mixed cultures may buffer the effects of the heatwave. Consequently, we also expect less fluctuation in the total abundance of prey bacteria in mixed cultures, as the presence of multiple flagellate strains feeding on the same prey should stabilize the predator–prey interaction and ensure functional redundancy in the face of environmental changes and potential shifts in the flagellate community.

## MATERIALS & METHODS

### Strains & cultivation

The strains *Pedospumella encystans* JBM/S11, *Spumella rivalis* AR4A6, and *Poteriospumella lacustris* JBM10 were previously isolated from soil or freshwater samples originating from different geographical locations (Table 1). The axenic strain JBM10 is routinely cultivated in NSY medium (nutrient broth, peptone from soybean, yeast extract; *Hahn et al., 2003*) in 100 ml Erlenmeyer flasks. Due to the lack of axenic strains associated with *Peodospumella* and *Spumella*, two xenic strains were used in this study. Both JBM/S11 and AR4A6 are routinely grown in inorganic basal medium (*Hahn et al., 2003*) in cell culture flasks (25 $cm^3$ with filter screw cap; TTP Techno Plastic Products AG) using wheat grains as a food source. The compositions of the NSY medium and the IB medium are provided in Table S1. All strains are grown in a climate chamber (SANYO Electric Co. Ltd., Osaka, Japan) at 15 °C under a 14h:10 h light-dark cycle. The bacterial strain *Linmohabitans* spp. IID5 is routinely grown as an axenic culture in NSY medium in 100 ml Erlenmeyer flasks at room temperature (RT) with constant shaking (96 rpm, orbital shaker; LAUDA-Brinkmann, LP, Marlton, NJ, USA).

### Experimental set-up with single-strain and mixed cultures

To create similar conditions for all three strains, all experiments were performed in IB medium. Therefore, flagellates were harvested by centrifugation for 10 min at 2,820 g at RT to remove the original medium and each resulting pellet was then resuspended in 50

**Table 1  Origin and affiliation of strains used in this study.**

| Strain | Species designation | 18S clade | Geographical Origin | Habitat origin | Media |
|---|---|---|---|---|---|
| JBMS11 | *Pedospumella encystans* | C1 | Austria, Mondsee, near Rauchhaus | soil | IB + grain of wheat |
| AR4A6 | *Spumella rivalis* | C2 | Austria, River Fuschler Ache | freshwater | IB + grain of wheat |
| JBM10 | *Poteriospumella lacustris* | C3 | Austria, Lake Mondsee | freshwater | NSY (3g/L) |

ml IB medium. One ml aliquots of a *Linmohabitans* spp. IID5 culture were centrifuged for 10 min at 15,000 g at RT and, after removal of the supernatant, resuspended in IB medium. One milliliter of the washed bacteria was added to each flagellate culture as food source. These pre-cultures were then incubated at 23 °C with a 14h:10 h light-dark cycle for three days to acclimatize to the new conditions. The bacteria from both xenic strains were isolated through filtration (diameter 25 mm, pore size 0.45 μm; Millipore GTTP 02500, Millipore, Eschborn, Germany) and cultivated in NSY medium in a 100 ml Erlenmeyer flask at 15 °C with a 14h:10 h light-dark cycle. All experiments were conducted in triplicates in 75 ml of IB medium in 100 ml Erlenmeyer flasks. Total flagellate abundance in the pre-cultures at the end of the acclimatization phase was counted in Sedgewick–Rafter chambers using 1 ml of the Lugol–fixed subsamples to adjust the flagellate abundance to approximately 30,000 cells per ml in the single-strain cultures. In the mixed cultures, the three strains were mixed to achieve relative abundances of 20% *Pedospumella*, 30% *Poteriospumella*, and 50% *Spumella*, mimicking the relative abundances of the respective clades in natural environments during late spring (*Weisse, 1997*; *Cleven & Weisse, 2001*; *Nolte et al., 2010*). The concentration of food bacteria was set to $2 \times 10^7$ bacteria per ml, consisting predominately of *Limnohabitans* spp. IID5 in addition to 10,000 cells per ml of the previously cultivated bacteria from the xenic flagellate cultures to ensure uniformed bacterial backgrounds in all cultures. Further, wheat grains were preheated at 70 °C for one hour and one wheat grain was added to each culture as food source for the bacteria. All cultures were initially incubated at 23 °C with a 14h:10 h light-dark cycle for two days. The temperature was then increased to 27 °C for three days before being reduced back to 23 °C for a two-day recovery phase. While 23 °C provides optimal growth conditions for representatives of all three clades, it has been shown that growth rates generally decrease at temperatures above 25 °C, particularly for strains associated with *Pedospumella* and *Spumella* (*Boenigk et al., 2007*). The 4 °C temperature increase exposed the cultures to conditions that induced heat stress, while also ensuring the survival of the cultures. Four ml subsamples were collected daily and fixed with 2% paraformaldehyde (PFA) for 1 h at RT or overnight at 4 °C and filtered onto white polycarbonate filters (diameter 25 mm, pore size 0.2 μm; Millipore GTTP 02500; Millipore, Eschborn, Germany). The filters were air-dried and stored at −20 °C for further use.

## Determination of cell counts and growth rates in single-strain and mixed cultures

Sections of the filters were cut out with a razor blade. To determine total flagellate abundance in single-strain cultures, the filter sections were stained with 4,6-diamidino-2-phenylindole (DAPI; 0.1 mg/ml) for 10 min at RT. To remove non-specific staining, the filters were washed in sterile distilled water for 1 min at RT, then dehydrated with 100% ethanol for 30 s at RT and air-dried. To determine the fraction of each strain in the mixed cultures, *Pedospumella encystans*, *Spumella rivalis*, and *Poteriospumella lacustris* were stained using fluorescently labelled clade-specific probes (O1C531, O2C613, O3C723; *Boden, Sieber & Boenigk, 2023*; 50 ng DNA/µL; Eurofins Genomics Germany GmbH, Ebersberg, Germany). Therefore, filter sections were incubated sequentially in 50%, 80%, and 100% ethanol solutions to dehydrate the samples. For each dehydration step, the filters were incubated for 3 min at RT. Then, the cells were hybridized with Cy3-labeled single-strand oligonucleotide probes as described by *Glöckner et al. (1996)*. To optimize signal detection rates, the concentration of SDS was increased to 0.02% in the hybridization buffer, and the hybridization time was extended to 3 h at 46 °C. To determine total flagellate abundance in the mixed cultures, the filter sections were afterward counterstained with DAPI as described before. After staining, the filter sections were mounted in the non-hardening and non-bleaching mounting medium CitiFluorTM AF2 (Citifluor, Ltd., London, United Kingdom), and images were captured using the NIS-Elements BR imaging software (Nikon Corp., Tokyo, Japan) and a Nikon Eclipse 80i microscope (Nikon Corp., Tokyo, Japan) with a 100x objective and appropriate fluorescence filters to detect DAPI and Cy3 signals. DAPI and Cy3 signals were counted manually using the software package Fiji (*Schindelin et al., 2012*).

## Data analysis

Growth rates ($\mu$) were calculated for three distinct phases throughout the experiment using R (v4.3.3; *R Core Team, 2024*) and RStudio (v2023.12.1; *RStudio Team, 2024*) and the formula

$$\mu = (\ln(C_t) - \ln(C_0))/t$$

where $C_t$ is the total flagellate abundances at the end of the respective interval and $C_0$ is the flagellate abundance at the beginning of the interval. All growth rates indicate growth per day. Specifically, growth rates were determined during the initial acclimatization phase from exponentially growing single strains as well as for all flagellates in the mixed cultures. Further, growth rates were assessed at the end of the heatwave, *i.e.,* between days five and six, and during the recovery phase, *i.e.,* between days seven and eight. Cell abundances, growth rates and the relative share of each taxa in the mixed cultures were plotted using R, RStudio and the R packages "readxl", "gglot", "cowplot" and "dplyr" (*Wickham & Bryan, 2023*; *Wickham, 2016*; *Wilke, 2024*; *Wickham et al., 2023*). Statistical data analysis was performed in R and RStudio. Growth rates were statistically compared among each other *via* Welch's ANOVA and Games-Howell *post-hoc* test using the R-packages "readxl", "dplyr" and "rstatix" (*Wickham & Bryan, 2023*; *Wickham et al., 2023*; *Kassambara, 2023*).

Similarly, the relative shares of *Pedospumella encystans*, *Spumella rivalis* and *Poteriospumella lacustris* in the mixed cultures at different sampling days were statistically compared *via* Welch's ANOVA and Games-Howell *post-hoc* test. Variability in bacterial abundances was statistically compared among all cultures *via* Brown-Forsythe test.

## RESULTS

After a lag phase between day one and day two, an increase in total flagellate abundance starting from day two was observed in all cultures (Fig. 1). Growth occurred between day two and day four in all cultures, with the exception of *Poteriospumella* single-strain experiments, where growth ceased by day three (Fig. 1). During this initial growth phase, hereafter referred to as "heatwave onset", significant differences among the cultures were observed when comparing flagellate growth in the single-strain cultures and in the mixed cultures (Welch's ANOVA; $p = 0.04$). Growth rates of the taxa were not significantly different when grown alone (Welch's ANOVA; $p = 1.00$; Fig. 2). But growth rates significantly differed during the heatwave onset (Welch's ANOVA; $p = 0.004$) between flagellates growing in single-strain culture and the same taxon growing in mixed culture: Growth rates for *Spumella* (Games-Howell test; $p = 0.02$) and *Pedospumella* (Games-Howell test; $p = 0.02$) were significantly lower in the mixed cultures compared to when they were grown individually. In contrast, growth rates of *Poteriospumella* did not differ between treatments (Games-Howell test; $p = 1.00$; Fig. 2). At the end of the heatwave total flagellate abundance decreased in all cultures (Fig. 1) and no significant differences were observed between the three taxa (Welch's ANOVA, $p = 0.98$; Fig. 2). Similarly, no significant differences were observed when comparing growth of the three taxa in single-strain culture and in mixed culture (Games-Howell test; *Pedospumella*: $p = 1.00$, *Spumella*: $p = 1.00$, *Poteriospumella*: $p = 1.00$). During the recovery phase no significant differences were observed in total flagellate growth when comparing single-strain experiments and mixed cultures (Welch's ANOVA; $p = 0.37$). Interestingly, the growth rates of *Poteriospumella* and *Spumella* in single-strain cultures were slightly but not significantly lower than in the mixed cultures (Games-Howell test; *Poteriospumella*: $p = 0.68$, *Spumella*: $p = 0.64$; Fig. 2). Replicates of the same treatment exhibited similar trends throughout the entire experiment (Fig. 2).

The relative share of all three taxa in the mixed culture (Fig. 3) changed significantly during the onset of the heatwave but was stable before and after: No significant changes were observed for any of the three taxa during the initial acclimatization phase, *i.e.,* between day one and day two (Games-Howell test; *Pedospumella*: $p = 1.00$, *Spumella*: $p = 0.20$, *Poteriospumella*: $p = 0.48$). During the onset of the heatwave, *i.e.,* between day two and three, the relative share of *Spumella* decreased significantly from 52% on day two to 34% on day three (Games-Howell test; $p < 0.001$). Similarly, the relative share of *Pedospumella* decreased significantly from 21% on day two to 11% on day three (Games-Howell test; $p < 0.001$). In contrast, the share of *Poteriospumella* increased significantly from 29% on day two to 55%, dominating the community on day three (Games-Howell test; $p < 0.001$). After this shift in community composition during onset of the heatwave, no further significant changes of the relative share were observed both during the heatwave (Games-Howell

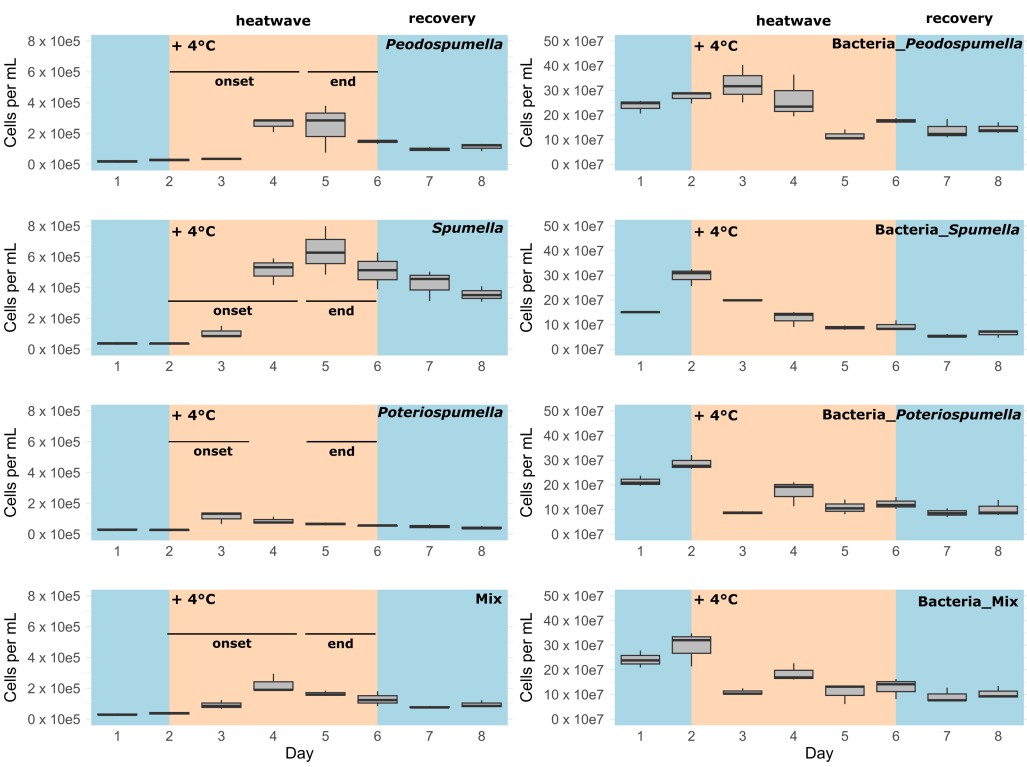

**Figure 1  Cell abundances under varying temperature conditions.** Total absolute abundance of flagellates (left) and bacteria (right) during phases of normal (blue) and increased (orange) temperature in single-stain cultures of (A) *Pedospumella encystans*, (B) *Spumella rivalis*, (C) *Poteriospumella lacustris* and (D) mixed cultures containing all three strains.

test; *Pedospumella*: $p = 0.97$, *Spumella*: $p = 0.99$, *Poteriospumella*: $p = 0.96$) and during the following recovery phase (Games-Howell test; *Pedospumella*: $p = 0.95$, *Spumella*: $p = 1.00$, *Poteriospumella*: $p = 1.00$). On day seven, *Poteriospumella* still dominated the community, constituting on average 56% of the total flagellate abundance, while *Pedospumella* and *Spumella* represented 12% and 34%, respectively, of all flagellates in the mixed cultures.

Furthermore, variability of the bacterial abundances over time differed significantly among the cultures (Brown-Forsythe test; $p = 0.01$). The relative variability in bacterial abundances was assessed by calculating the coefficient of variation (CV) for each culture. *Pedospumella* single-strain cultures exhibited the lowest CV at 39.1%, indicating the least variability in bacterial abundance throughout the experiment. In contrast, *Spumella* single-strain cultures showed the highest CV at 58.8%. *Poteriospumella* single-strain cultures and mixed cultures displayed similar CVs of 49.4% and 49.6%, respectively (Fig. S1).

## DISCUSSION

The impact of temperature shifts, *e.g.*, due to global warming and heatwaves, has been the subject of numerous studies (*Font et al., 2021*; *Woolway et al., 2021*), revealing substantial shifts in protist communities (*Hao et al., 2018*; *Thomson & Manoylov, 2019*). We observed

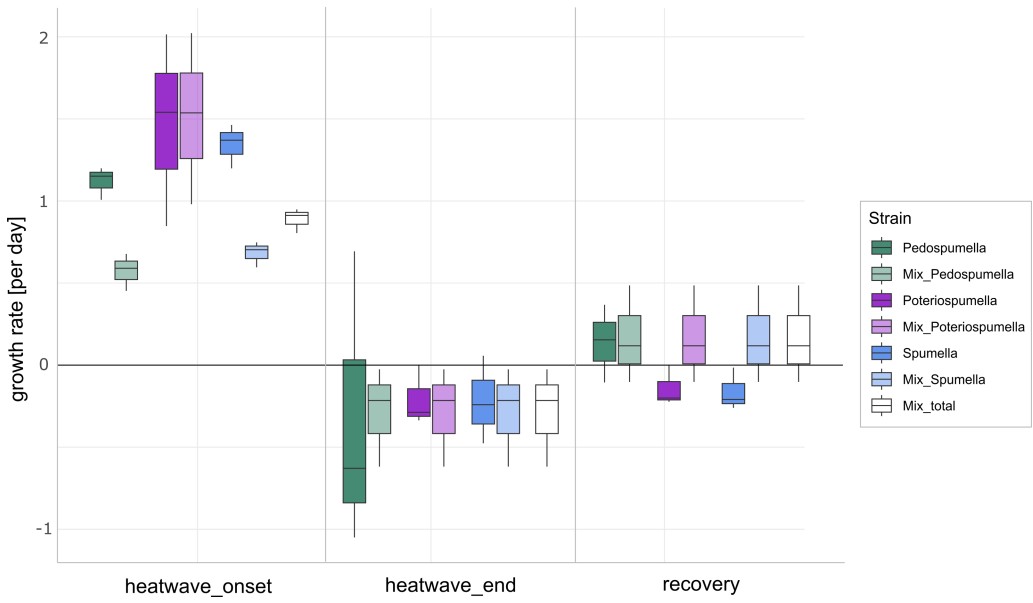

**Figure 2  Flagellate growth in single-strain and mixed cultures.** Growth rates [day$^{-1}$] of *Pedospumella encystans*, *Spumella rivalis* and *Poteriospumella* lacustris in single-strain cultures and in mixed cultures as well as total flagellate growth in the mixed cultures during the onset of the heatwave, at the end of the heatwave and during the recovery phase. Negative growth rates correspond to mortality rates.

significant differences both among the different cultures and when comparing growth of each taxa in single-strain and mixed cultures (Fig. 2). While total flagellate growth was generally higher in single-strain experiments than in the mixed cultures, *Poteriospumella* exhibited the highest growth rate among all cultures. Interestingly, similar growth rates were observed for *Poteriospumella* in both single-strain and mixed cultures, contrasting with *Pedospumella* and *Spumella*, which displayed significantly lower growth rates in mixed cultures compared to when they were grown individually (Fig. 2). This is in accordance with previous research that identified *Poteriospumella*'s competitive advantage, particularly under elevated temperature conditions (*Boden, Sieber & Boenigk, 2023*). Additionally, we observed a shift in the community composition in the mixed cultures. At the beginning of the experiment, the community was dominated by *Spumella* but its relative share decreased significantly during the onset of the heatwave. In contrast, the relative abundance of *Poteriospumella* increased significantly, becoming the dominant taxa approximately 24 h after the temperature rise (Fig. 3). This shift in taxon composition in our mixed cultures is in accordance with seasonal environmental data that identified similar temperature-induced shifts in a natural lake, with *Spumella* typically dominating in winter and *Poteriospumella* in summer (*Nolte et al., 2010*). No shifts in community composition were observed in the mixed cultures at the end of the heatwave and during the recovery phase, *i.e.,* the community composition remained unchanged despite changing temperature conditions (Fig. 3). This was unexpected as the influence of abiotic factors such as temperature, salinity or oxygen availability on flagellate abundance and community composition has been demonstrated in previous studies (*Nolte et al., 2010*; *Princiotta & Sanders, 2017*;
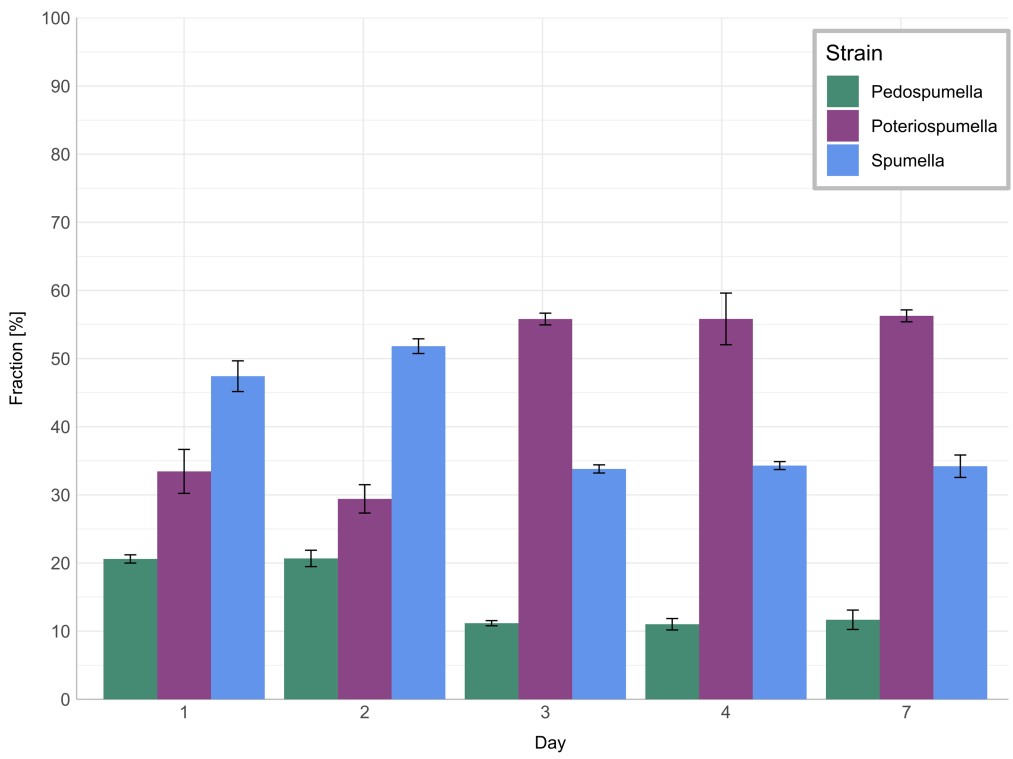

**Figure 3  Community composition in mixed cultures.** Relative shares of *Pedospumella encystans*, *Poteriospumella lacustris* and *Spumella rivalis* in the mixed cultures. The bars indicate the mean percentage of each strain in all mixed cultures on day 1, day 2, day 3, day 4, and day 7. Error bars represent standard deviations.

*Gran-Stadniczeňko et al., 2018*). However, most of these studies were investigating the effects of abiotic factor in environmental samples, indicating that mock communities in laboratory experiments may be less dynamic than natural communities, possibly due to reduced competition in species-poor communities. At the end of the heatwave, the growth rates of *Poteriospumella*, both in single-strain and mixed cultures, were similar to those of *Pedospumella* and *Spumella* and no competitive advantage was identified for any of the three taxa during that later phase. The reasons why *Poteriospumella* failed to maintain its competitive advantage throughout the entire duration of the heatwave remain unclear. All cultures reached maximal abundance latest by day five, therefore potential reasons include density-dependent factors like the accumulation of toxic metabolic byproducts such as acids and alcohols, or oxygen depletion, which are common phenomena in cultures with high cell density and would affect all three taxa equally (*Rouf et al., 2017*). Additionally, the daily sampling resulted in a reduction of medium in all cultures which could further impact flagellate growth particularly during the latter half of the experiment. Beside abiotic factors, biotic interactions can also influence flagellate growth in the cultures. High cell densities can also lead to cannibalism among flagellates, particularly when alternative prey is scarce or competition is high (*Martel & Flynn, 2008*; *Yang et al., 2020*). However, we

found no indications of cannibalism in our samples, likely because the concentration of prey bacteria remained above $5 \times 10^6$ cells per milliliter throughout the experiment, which was sufficiently high to prevent starvation (*Boenigk et al., 2002*). Given the consistently high concentration of prey bacteria, it is unlikely that nutrient depletion, a common cause of mortality in dense microbial cultures (*Rouf et al., 2017*), was responsible for the decline in total flagellate abundances at the end of the heatwave. Our results indicate that the abiotic and/or biotic density-dependent factors arising from the experimental setup overshadowed the effects of the temperature increase during the latter half of the heatwave. While heterotrophic nanoflagellates can reach abundances of several thousand cells per milliliter in natural habitats, their total flagellate abundances remain notably lower than the maximum abundances that were reached in this study (*Boenigk & Arndt, 2002*). Therefore, the response to the temperature increase that was observed during the onset of the heatwave may be a closer representation of natural habitats than towards the end of the heatwave. In the recovery phase, growth rates of *Spumella* and *Poteriospumella* were slightly but not significantly higher in the mixed cultures than in single-strain cultures, indicating that the lower total flagellate abundance during the recovery phase may reduce the impact of density-dependent factors, thereby allowing temperature-related effects to become dominant again.

As important players in the top-down control of bacterial populations, changes in both the total abundance and community composition of heterotrophic nanoflagellates significantly influence the dynamics of bacterial prey populations (*Sherr & Sherr, 2002*). In this study, variability in the bacterial abundances differed among the single strain-cultures, with the highest variation observed in *Spumella* single-strain cultures. Notably, within mixed cultures, bacterial abundances displayed similar variability than in the *Poteriospumella* single-strain cultures. The strong fluctuations observed in *Spumella* single-strain cultures were likely mitigated in the mixed cultures by the presence and dominance of *Poteriospumella*. These results further support the hypothesis that functional redundancy, demonstrated here by the simultaneous presence of taxa capable of feeding on the same prey but exhibiting different adaptations to abiotic factors, can stabilize habitats when confronted with changing environmental conditions. Interestingly, variability of bacterial abundance was lowest in *Pedospumella* single-strain cultures. However, *Pedospumella* showed low competitive strength in the presence of the other two taxa, thus failing to further stabilize bacterial populations in the mixed cultures. This highlights the importance of biotic interactions when assessing the impact of extreme climate events on natural microbial communities.

## CONCLUSIONS

Our findings highlight the complexities of protist community dynamics under thermal stress, indicating that while temperature can drive significant changes in abundance and composition of flagellate communities, other factors such as density-dependent interactions and prey availability also significantly impact microbial communities. Our study indicates that stabilizing effects due to functional redundancy may be pronounced as long as

density-dependent factors play a minor role. This study underscores the importance of considering both abiotic and biotic factors when assessing the impacts of climate change on microbial communities, as well as the potential for functional redundancy to stabilize ecosystems facing environmental fluctuations.

### Funding
This study was performed within the Collaborative Research Center (CRC) RESIST and analyses were mainly done in Project A06, funded by the German Research Foundation (DFG) –CRC 1439/1; project number INST 20876/402–1. The funders had no role in study design, data collection and analysis, decision to publish, or preparation of the manuscript.

### Grant Disclosures
The following grant information was disclosed by the authors:
Collaborative Research Center (CRC) RESIST.
German Research Foundation (DFG)- CRC: 1439/1, INST 20876/402-1.

### Competing Interests
The authors declare there are no competing interests.

### Author Contributions
- Lisa Boden performed the experiments, analyzed the data, prepared figures and/or tables, authored or reviewed drafts of the article, and approved the final draft.
- Chantal Klagus performed the experiments, analyzed the data, authored or reviewed drafts of the article, and approved the final draft.
- Jens Boenigk conceived and designed the experiments, authored or reviewed drafts of the article, and approved the final draft.

### Data Availability
Raw data are available in the Supplemental Files.

### Supplemental Information
Supplemental information for this article can be found online at http://dx.doi.org/10.7717/peerj.17912#supplemental-information.

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
