# Peer review of "Insights from single-strain and mixed culture experiments on the effects of heatwaves on freshwater flagellates"

_PeerJ, doi:10.7717/peerj.17912_

## Round 0.1 · original submission · Minor Revisions

The paper requires some moderate revision. Please provide a detailed point-by-point rebuttal letter to each of the reviewers' comments, along with your revised manuscript.

Reviewer 1 ·

Basic reporting

no comment

Experimental design

no comment

Validity of the findings

no comment

Additional comments

See attached PDF

Annotated reviews are not available for download in order to protect the identity of reviewers who chose to remain anonymous.

·

Basic reporting

This is a well-structured manuscript, with clear, literature-driven hypotheses and suitable experimental design to address them.

Experimental design

The experimental set-up is restricted to three morphologically similar heterotrophic nanoflagellates, but can be replicated and tested across phylogenetic groups, providing broader relevance and ecological interest, especially under the prism of climate change acute impacts on aquatic systems.

Specific comments:
L148-150: What was the volume of the subsamples? Can you comment on the total volume decrease of the cultures across the 8-day duration of the experiment because of samplings? Can this stress/affect flagellate dynamics and predator-prey relationship?
L175-180: Please clarify how were growth rates calculated and the measurement units. E.g. (Abundance in t1- abundance in t0)/Abundance in t0) in growth per day?

Validity of the findings

Specific comments:

Results: I would like a sentence indicating if there were significant differences among replicates of the same treatment. I assume not, but I think it should be mentioned.
L202-204: Except for Poteriospumella.
Figures 1 & 2 and respective results: My understanding from reading figure 1 is that flagellates’ abundances in both single-strain and mixed cultures more or less increased during the heatwaves in all cases, compared to the acclimation stage. However, in figure 2, a high loss rate (negative growth values) is indicative in all cases, I assume because of examination of consecutive days, Thus, in the end of the heatwave period there is decrease which affects growth rate values, but overall, the abundance within the heatwave days is higher. Maybe, it would be more informative to additionally show growth rates comparing temperature stages of the cultures, rather than differences between days?
L234-240: Since these results correspond to the last hypothesis of the manuscript (L108-112), which is very relevant and interesting, I suggest that the addition of a graph showing these results would improve understanding. As far as I understand, these data -showing relatively high variations of bacterial abundances in mixed cultures- reject the hypothesis of mixed cultures stabilizing predator-prey interactions, which is a very interesting discussion. This discussion is somewhat restricted (and not in exact accordance with the data) in the manuscript (L298-306).

---

## Round 0.2 · accepted · Accept

Thank you for your detailed responses, the article is now Acceptable.